# Metformin hydrolase is a recently evolved nickel-dependent heteromeric ureohydrolase

M. Sinn ®[1] ✉, L. Riede[1], J. R. Fleming[2], D. Funck ®[1], H. Lutz ®[2], A. Bachmann[2], O. Mayans[2,3] & J. S. Hartig ®[1,3] ✉

The anti-diabetic drug metformin is one of the most widely prescribed medicines in the world. Together with its degradation product guanylurea, it is a major pharmaceutical pollutant in wastewater treatment plants and surface waters. An operon comprising two genes of the ureohydrolase family in *Pseudomonas* and *Aminobacter* species has recently been implicated in metformin degradation. However, the corresponding proteins have not been characterized. Here we show that these genes encode a Ni$^{2+}$-dependent enzyme that efficiently and specifically hydrolyzes metformin to guanylurea and dimethylamine. The active enzyme is a heteromeric complex of α- and β- subunits in which only the α-subunits contain the conserved His and Asp residues for the coordination of two Ni$^{2+}$ ions in the active site. A crystal structure of metformin hydrolase reveals an $α_2β_4$ stoichiometry of the hexameric complex, which is unprecedented in the ureohydrolase family. By studying a closely related but more widely distributed enzyme, we find that the putative predecessor specifically hydrolyzes dimethylguanidine instead of metformin. Our findings establish the molecular basis for metformin hydrolysis to guanylurea as the primary pathway for metformin biodegradation and provide insight into the recent evolution of ureohydrolase family proteins in response to an anthropogenic compound.

About 10% of the adult population worldwide suffered from diabetes in 2021 with increasing tendency, causing enormous costs for medical treatment[1]. For the strongly prevailing type II diabetes, metformin is currently the most efficient treatment[2]. With an average daily dose of 2 g per day and patient, metformin is one of the most-produced and consumed drugs worldwide[3,4]. Metformin acts on multiple, not fully understood targets in humans with the main effect of inhibiting gluconeogenesis in liver mitochondria[5,6]. Additionally, metformin has been discussed controversially to have beneficial effects in several other diseases. It has a half-life of ~5 h in humans and is excreted unchanged from the body[7]. Metformin, as well as its major bacterial

metabolite guanylurea, is not efficiently removed by wastewater treatment plants but is released into surface waters, where it is currently one of the major pharmaceutical pollutants[4,8–10]. With few exceptions, metformin and guanylurea have so far not been found to be harmful in the environment at concentrations found in surface waters[11]. Nevertheless, increased prescriptions and continued accumulation could lead to increased metformin concentrations in the biosphere. Bioremediation is a promising strategy for removing anthropogenic compounds from the environment before they reach harmful concentrations. Microbial communities have been shown to degrade metformin to guanylurea and further[12,13], but these pathways

[1]Department of Chemistry, University of Konstanz, Konstanz, Germany. [2]Department of Biology, University of Konstanz, Konstanz, Germany. [3]Konstanz Research School Chemical Biology (KoRS-CB), University of Konstanz, Konstanz, Germany. ✉e-mail: malte.sinn@uni-konstanz.de; joerg.hartig@uni-konstanz.de

are only beginning to be understood at the molecular level. Tassoulas et al. isolated *Pseudomonas mendocina* GU using guanylurea as the sole nitrogen source. They showed that guanylurea is hydrolyzed to guanidine and ammonia by the isochorismate hydrolase-like protein GuuH and then completely assimilated by the guanidine carboxylase pathway[14–16].

The enzyme catalyzing the hydrolysis of metformin to guanylurea remains unknown. After an initial report on *Pseudomonas* strains able to use metformin as nitrogen source[17], recently, three independent studies characterized *Aminobacter* and *Pseudomonas* strains from sewage sludge that were able to grow on metformin as sole carbon and nitrogen source[18–20]. All three teams identified a group of genes shared among the isolated strains that likely encodes an enzyme system catalyzing the hydrolysis of metformin to guanylurea and dimethylamine. These genes were found on the chromosome in two *Aminobacter* strains[18,19] but were found located on plasmids in a further *Aminobacter* strain and several *Pseudomonas* isolates[20]. Among the genes shared between the metformin-degrading strains are two nickel chaperone genes *hypA* and *hypB*. They are followed by two genes annotated as encoding ureohydrolase family proteins that are the best candidates for metformin hydrolase enzymes. The ureohydrolase family proteins are homologs of the $Ni^{2+}$-dependent guanidine hydrolase GdmH that was also found to be activated by nickel chaperones[21]. Directly downstream of the four putative metformin degradation genes, a homolog of the nickel uptake transporter HupE is located. Further genes in the chromosomal *Aminobacter* operon are related to urea transport and metabolism.

We heterologously expressed the candidate proteins of *Aminobacter niigataensis* MD1[18] and *Pseudomonas mendocina* MET[20] and purified an active hydrolase that efficiently and specifically converted metformin to guanylurea and dimethylamine. We solved the crystal structure of the metformin hydrolase from *A. niigataensis* MD1 that revealed the formation of a heterohexamer with $\alpha_2\beta_4$ stoichiometry in which only the α-subunits contained catalytically active di-nickel metal centers. During the course of revision Tassoulas et al.[22] and Li et al.[23] reported studies on the molecular mechanism of MefH that largely confirm the results presented here. Furthermore, we show that the highly similar operon of the *Aminobacter niigataensis* type strain (DSM7050) that was not able to grow on metformin[18] and a more distantly related operon from *Hyphomicrobiales* bacteria encode dimethylguanidine hydrolases without detectable metformin hydrolase activity. We propose that this chromosome-encoded activity is the evolutionary predecessor of the metformin hydrolase found in the bacterial isolates that assimilate metformin.

## Results

### The tandem ureohydrolase genes encode a metformin hydrolase

We heterologously expressed a part of the operon associated with metformin degradation containing both ureohydrolase family genes and the nickel chaperones *hypA* and *hypB* but without the nickel transporter *hupE* of *P. mendocina* MET, *A. niigataensis* MD1 and *A. niigataensis* DSM7050 in *E. coli* (Fig. 1A). For purification, the first of the ureohydrolase family proteins contained an N-terminal 6x His- or Strep-tag (for the different expression constructs used for the enzyme production see Supplementary Fig. 1). Strikingly, two bands were observed on denaturing polyacrylamide gels after affinity purification, matching the predicted sizes of the first and second ureohydrolase family proteins (Fig. 1C; Supplementary Fig. 2). Peptide mass fingerprint analysis confirmed that the untagged second ureohydrolase family proteins were co-purified together with the tagged proteins (Supplementary Table 1). To avoid co-purification of naturally His-rich HypB and interference between the Ni-NTA matrix

used for purification and the putatively nickel-dependent ureohydrolases, we mostly used the Strep-tagged versions of the first ureohydrolase for the enzymatic characterization. Key experiments were also performed with the 6x His-tagged versions and yielded very similar results.

Incubation of metformin with the purified ureohydrolases from *P. mendocina* MET and *A. niigataensis* MD1 yielded guanylurea and dimethylamine (Fig. 1B), which were detected by LC-MS. We tested the specificity of the enzymes by incubation with several other guanidine compounds as potential substrates (Supplementary Table 2). Of all tested compounds only metformin and dimethylguanidine were hydrolyzed. For all other compounds, including highly similar methylguanidine, neither consumption of the substrate nor product formation were detected. The hydrolysis of dimethylguanidine produced dimethylamine and urea. Since urea is not ionized during LC-MS analysis, it was detected by a colorimetric assay[24] that was also used in the quantitative analyses of enzyme activity. We developed a quantitative method to detect guanylurea by LC-MS (Supplementary Fig. 3, see Methods section for details) and determined the specific metformin and dimethylguanidine hydrolysis activities of the three different enzymes (Fig. 1D). The enzymes of *P. mendocina* MET and *A. niigataensis* MD1 catalyzed the hydrolysis of both metformin and dimethylguanidine. However, the specific activity for dimethylguanidine hydrolysis was approximately 20-fold lower (Fig. 1D). It has been noted before that the type strain *A. niigataensis* DSM 7050 was not able to grow on metformin as the sole carbon source[18] despite possessing a highly similar operon (~93% identical amino acids for both the first and second ureohydrolase family proteins (Supplementary Fig. 4). Indeed, the enzyme of *A. niigataensis* DSM 7050 only hydrolyzed dimethylguanidine but not metformin (Fig. 1D). In the following, we refer to the gene products from the metformin-degrading strains as metformin hydrolase α and β (MefHα and MefHβ) and the proteins from *A. niigataensis* DSM 7050 as dimethylguanidine hydrolase α and β (DmgHα and DmgHβ). When *Pm*MefHα or *Pm*MefHβ were expressed as individual proteins in combination with the nickel chaperones, the purified proteins did not exhibit metformin hydrolase activity and the major part of *Pm*MefHα formed insoluble inclusion bodies (Supplementary Fig. 5). Therefore, we concluded that only heteromeric complexes of both subunits are catalytically active. MefH and DmgH are used throughout the manuscript to refer to the enzymatically active heteromeric complexes.

### Enzyme characterization

The presence of the nickel chaperones *hypA* and *hypB* in all three operons strongly suggested that the enzymes are dependent on nickel. *Pm*MefH was expressed in the presence of $Ni^{2+}$, $Mn^{2+}$, $Zn^{2+}$ and $Co^{2+}$ that have been reported as metal co-factors for ureohydrolases. Subsequently, the enzyme was purified (Supplementary Fig. 5) and the specific activity with 2 mM metformin was determined. The enzyme with the highest activity was obtained when expression occurred in the presence of $Ni^{2+}$ with only residual activity for the other metal ions (Fig. 1E). Consistent results were obtained for DmgH and dimethylguanidine as substrate (Supplementary Fig. 6). Therefore, we conclude that both MefH and DmgH are indeed nickel-dependent enzymes. Active preparations of DmgH expressed in the presence of $Ni^{2+}$ were subjected to ICP-OES analysis to determine the metal content. The nickel content was 3.0 μmol (μmol protein)$^{-1}$ and additionally 0.5 μmol (μmol protein)$^{-1}$ manganese were detected, but no other transition metals (Supplementary Fig. 7).

After establishing that the enzymes are dependent on nickel, the kinetic parameters were determined. Enzymes were expressed in the presence of $Ni^{2+}$ and purified by Strep-Tactin affinity chromatography. Purified enzymes were incubated with 0.16–100 mM of their preferred substrate (Fig. 1F). $K_M$ values for the enzymes were very similar with $61.9 \pm 4.4$ mM, $57.8 \pm 4.3$ mM and $57.1 \pm 4.3$ mM for *Pm*MefH, *An*MefH

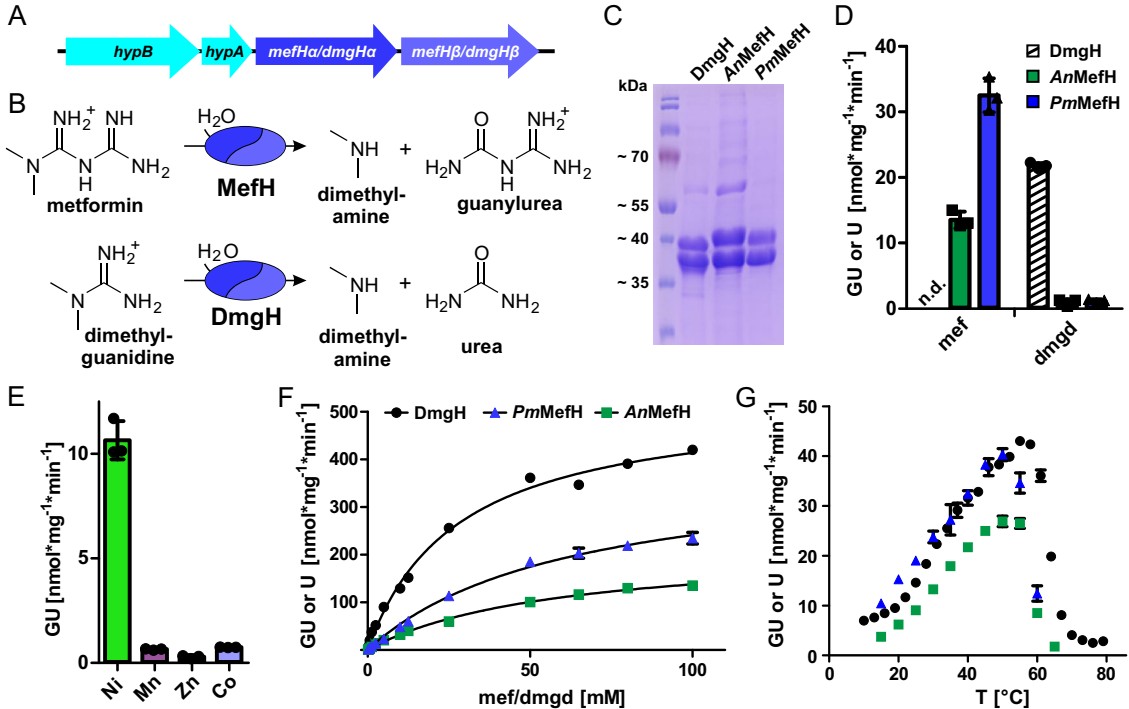

**Fig. 1 | Purification and activities of recombinant metformin and dimethylguanidine hydrolases MefH and DmgH. A** Genomic structure of the genes conferring metformin or dimethylguanidine hydrolase activity. **B** Reactions catalyzed by heteromeric metformin hydrolase MefH and dimethylguanidine hydrolase DmgH. **C** Representative Coomassie stained denaturing polyacrylamide gel of DmgH, *A. niigataenis An*MefH and *P. mendocina Pm*MefH purified by Strep-Tactin affinity chromatography after expression in the presence of 0.5 mM Ni$^{2+}$. In all three expression systems, the first hydrolase was Strep-tagged and the second hydrolase was co-purified, indicating the formation of a heteromeric complex. **D** Bar chart showing the specific activities with 2 mM metformin or dimethylguanidine as substrates of the three different enzyme systems. Activities were determined as the substrate and enzyme-dependent production of urea (U) or guanylurea (GU). Guanylurea was determined by LC-MS and urea was determined by a colorimetric assay. **E** Metal dependency of *Pm*MefH. *Pm*MefH was expressed in the presence of 50 μM of the respective metal ions and purified by nickel affinity chromatography

(Supplementary Fig. 5). Guanylurea production rates were determined for purified *Pm*MefH expressed in media supplemented with the divalent cations of the indicated metals. **F** Urea (U) or guanylurea (GU) production at different substrate concentrations was measured to determine Michaelis constant $K_M$ and maximal reaction rate ($v_{max}$) for DmgH (black circles), *An*MefH (green squares) and *Pm*MefH (blue triangles). Data were fitted using the Michaelis-Menten equation ($R^2$ = 0.9934; $R^2$ = 0.9946, $R^2$ = 0.9950, respectively). **G** Urea (U) or guanylurea (GU) production was measured at different temperatures to determine the apparent temperature optimum of MefH and DmgH. Legend as in **E**. For **D–G**, the data represent the average of triplicates and consistent results were obtained with independent preparations ($n$ = 3, error bars, s.d.). Please note that for the temperature dependency of DmgH in **G** the mean of duplicates are shown and some error bars are not visible in **F** and **G** as they are smaller than the symbols used to indicate the means. Source data are provided as a Source Data file.

and DmgH, respectively. Based on the observation of two catalytic sites per hexameric holoenzyme (see below), we calculated the turnover numbers as $k_{cat}$ = 0.9 s$^{-1}$ for DmgH and 0.8 s$^{-1}$ for *Pm*MefH but only 0.4 s$^{-1}$ for *An*MefH. Therefore, hydrolysis by DmgH and *Pm*MefH was more efficient than by *An*MefH. During revision of this manuscript, Tassoulas et al. reported that Tris buffer inhibited the enzyme[22]. We therefore repeated the purification and analysis of *An*MefH and DmgH with HEPES instead of Tris buffer. In HEPES, lower $K_M$ values were obtained with 1.9 ± 0.2 and 2.6 ± 0.1 mM, respectively (Supplementary Fig. 8). The $k_{cat}$ values were not affected by the buffer. All enzymes were highly resistant to thermal inactivation. DmgH exhibited a slightly higher apparent temperature optimum than *An*MefH and *Pm*MefH around 56 °C compared to 50 °C, respectively (Fig. 1G).

## Crystal structure of MefH
To determine the stoichiometry and architecture of the MefH heteromer, we resolved its three-dimensional atomic structure using X-ray crystallography. For crystallization, we used *An*MefH as it shares a higher sequence identity with DmgH than *Pm*MefH (93.5% and 93.3% sequence identity for the alpha subunits, respectively), so that residues responsible for the change in substrate specificity are easier to identify. The best diffraction data was obtained from *An*MefH crystals that were soaked with urea prior to vitrification and X-ray analysis. In the

obtained structure, both MefHα and MefHβ adopt an almost identical three-layer alpha-beta-alpha fold (Fig. 2A, B) typical for members of the ureohydrolase family[25], but diverge notably at the N-terminal loop regions that are distinct to each subunit type (Fig. 2C). The global structure of MefH is distinct amongst ureohydrolase family proteins since it is a heterohexamer consisting of a dimer of heterotrimers, each with αβ$_2$ stoichiometry (Fig. 2). From this stoichiometry, a molecular weight of 236 kDa is predicted for the recombinant MefH. In analytical size exclusion chromatography, a homogenous species with estimated size of 196 kDa was observed (Supplementary Fig. 9A). Subsequent dynamic light scattering measurement confirmed a monodisperse solution of particles with 10 nm diameter, which is in agreement with the size of the hexamer in the crystal structure (Supplementary Fig. 9B). The trimers interact with each other generating a two-fold symmetry axis along the enzyme. The MefHα subunits are located in apical position and make a close contact between both trimers. In contrast, the MefHβ subunits interact weakly with their counterparts in the opposite trimer. They contain flexible and partially disordered loops at the interface that result in the presence of a cleft between the trimers (Fig. 2B). Each MefHα subunit coordinates two metal ions in its active site through four Asp (D$_{183}$, D$_{187}$, D$_{276}$ and D$_{278}$) and two His (H$_{158}$ and H$_{185}$) residues. The electron density in the active site is best explained by two Ni$^{2+}$ ions, which is in agreement with the ICP-OES

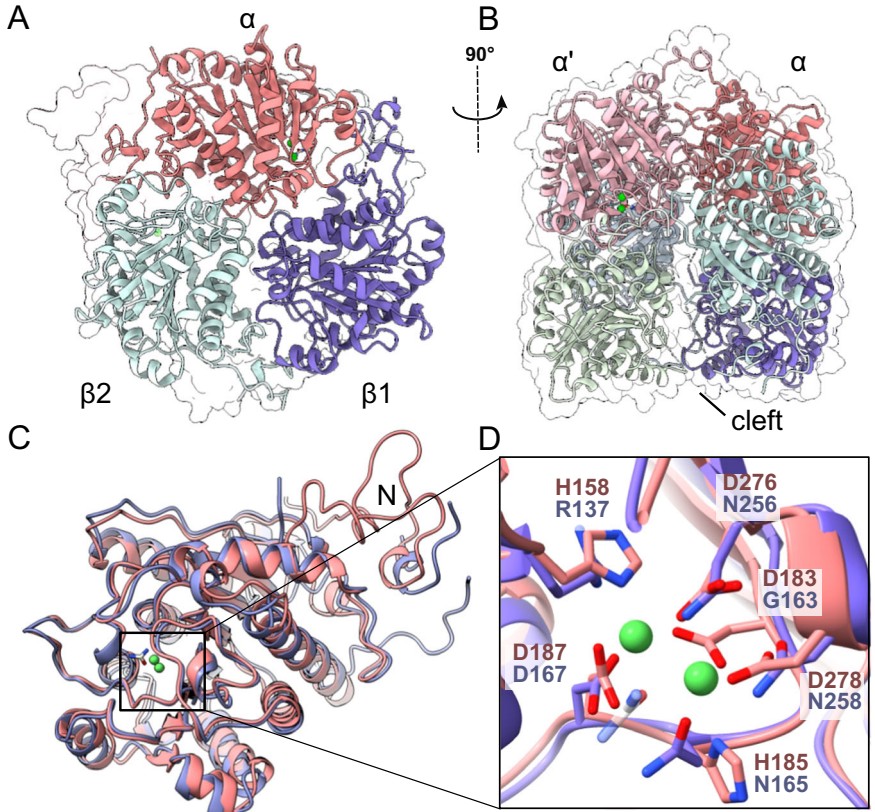

**Fig. 2 | Crystal structure of MefH. Global view of the MefH crystal structure.** MefH is a dimer of two heterotrimers. Green spheres depict the $Ni^{2+}$ ions in the active site of the MefHα subunits. **A** The heterotrimer is formed by the active MefHα subunit (red) and two inactive MefHβ subunits (blue). **B** The MefHα subunits interact tightly whereas between the loosely interacting MefHβ subunits a cleft is formed. **C** Overlay of a MefHα (red) and a MefHβ (purple) subunit with bound $Ni^{2+}$ (green spheres) and urea (beige). **D** Close-up of the active site of MefHα with the side chains of the nickel-coordinating amino acids and the corresponding amino acids of the degenerated binding site of MefHβ, where no metal was bound. For visual clarity, biuret bound at the N-termini of the MefHα subunits is not displayed. Ligand binding is further illustrated in Supplementary Figs. 10, 11. Model coordinates and experimental diffraction data have been deposited with the Protein Data Bank under accession code 8RYI.

experiments and the metal dependency of the enzyme. However, we cannot exclude that $Ni^{2+}$ is replaced by $Mn^{2+}$ in a small fraction of the binding sites. In the crystal structure, the $Ni^{2+}$ ions coordinate a molecule of urea (Supplementary Fig. 10). In contrast, the corresponding region of the MefHβ subunits do not contain any metal ions or ligands, and four of the six residues that bind $Ni^{2+}$ in the MefHα subunits are replaced by amino acids that are not prone to coordinate cations (Fig. 2D). Another specific feature of the MefHα subunits is the binding of one molecule per subunit in the N-terminal loop segment that was interpreted as biuret (Supplementary Fig. 11). The electron density could also satisfactorily be interpreted as guanylurea. However, neither metformin nor guanylurea were present in the crystallization solutions, whereas biuret is a known contaminant of urea that was used for soaking. The biuret binding site is distant from the di-metal center marking the active site and it is unknown whether it might hold functional significance.

## Investigation of substrate specificity determinants

To investigate the similarity between the different enzyme systems and subunits, we constructed a multiple sequence alignment of all six subunits of both MefH and DmgH enzymes (Supplementary Fig. 4). AnMefHα and DmgHα share 93.5% sequence identity, whereas AnMefHα and AnMefHβ share only 33% sequence identity. MefHβ and DmgHβ exhibit 93% sequence identity. Thus, the protein subunits within one heteromeric enzyme are more dissimilar than the corresponding subunits of the two distinct enzymes. AnMefHα and PmMefHα share a sequence identity of 97.5% and their β-subunits are

100% conserved. As observed in the crystal structure, the metal coordination site is degenerated in MefHβ and DmgHβ. A phylogenetic analysis including further ureohydrolase sequences with >24% sequence identity to MefHα suggests that DmgH is likely the direct evolutionary precursor of MefH (Supplementary Fig. 12). The α and β subunits of homologous proteins form separate branches in the tree and are clearly separated from other ureohydrolase family proteins that are encoded by single genes. A second tandem arrangement of ureohydrolase genes in putative Limnocylindrales bacteria seems to have evolved independently. To support our hypothesis, we investigated the DmgH homolog from Hyphomicrobium, with 70 % sequence identity to DmgHα and 68% sequence identity to DmgHβ. The Hyphomicrobium DmgH hydrolyzed dimethylguanidine with comparable specific activities as DmgH from A. niigataensis but did not hydrolyze metformin (Supplementary Fig. 13). Taken that a homolog of DmgH with lower sequence identity compared to MefH exhibits the same substrate specificity and the high sequence identity between DmgH and MefH, we propose that DmgH is the evolutionary precursor of MefH.

In order to investigate the contribution of the subunits to the substrate specificities of MefH and DmgH, we cloned constructs for the co-expression of MefHα with DmgHβ and DmgHα with MefHβ. Both combinations yielded stable heteromers and DmgHα/MefHβ exclusively hydrolyzed dimethylguanidine, whereas MefHα/DmgHβ preferentially hydrolyzed metformin (Fig. 3B and Supplementary Fig. 14). However, the specific activity of DmgHα/MefHβ was approximately 5 times lower than for native DmgH, whereas

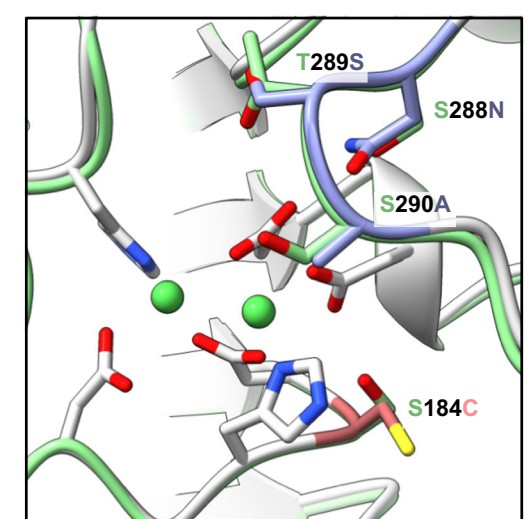

**A**

**B**

| variant | specific activity (nmol mg$^{-1}$ min$^{-1}$) | |
| --- | --- | --- |
| | dimethylguanidine | metformin |
| MefHα / DmgHß | 0.4 ± 0.1 | 14.5 ±1.1 |
| DmgHα / MefHß | 3.3 ± 0.1 | n.d. |
| native DmgH | 21.8 ± 0.4 | n.d. |
| DmgHα(S$_{288}$N) / DmgHß | 2.8 ± 0.3 | n.d. |
| DmgHα(T$_{289}$S) / DmgHß | 50.0 ± 0.9 | n.d. |
| DmgHα(S$_{290}$A) / DmgHß | 0.75 ± 0.04 | n.d. |
| DmgHα(STS→NSA) DmgHß | 0.4 ± 0.1 | n.d. |
| DmgHα(STS→NSA, S$_{184}$C) / DmgHß | 0.7 ± 0.3 | 0.19 ± 0.01 |

**Fig. 3 | Investigation of substrate specificity determinants. A** Overlay of the active sites of the *An*MefH crystal structure (gray) and an AlphaFold model of DmgH (green). The side chains of the metal binding residues of *An*MefH are shown (gray). These amino acids are identical in DmgH and MefH. Residues that were exchanged in DmgH for the corresponding residues of MefH are highlighted in blue, if substrate specificity was not changed, or red, if substrate specificity was changed. **B** Specific activities for the combinations of DmgH and *An*MefH subunits and different variants of DmgH with 2 mM dimethylguanidine or metformin. Data represent the average ± s.d. of triplicates and consistent results were obtained with independent preparations (*n* = 3, error, s.d.). n.d. not detected. Source data are provided as a Source Data file.

MefHα/DmgHβ was less affected in comparison to MefH. These results clearly show that the catalytic activity and the substrate selectivity is mainly mediated by the MefHα and DmgHα subunits. The degenerated subunits MefHβ and DmgHβ were crucial for activity, but were exchangeable and had only a minor effect on substrate specificity.

As the proteins are very similar with only 23 amino acid residues distinguishing MefHα from DmgHα, we decided to investigate variants of DmgHα for changed substrate specificity. Especially, a stretch of three amino acids in close vicinity to the two Ni$^{2+}$ ions in the active site aroused our interest. Instead of Ser$_{288}$-Thr$_{289}$-Ser$_{290}$ in DmgHα, these positions are exchanged for Asn$_{288}$-Ser$_{289}$-Ala$_{290}$ in *An*MefHα and *Pm*MefHα (Fig. 3A and Supplementary Fig. 4). We mutated all three single positions and the three amino acid stretch in the DmgH expression construct and determined the specific activities of the resulting protein variants (Fig. 3B and Supplementary Fig. 14). Remarkably, the Thr$_{289}$Ser exchange resulted in higher dimethylguanidine hydrolase activity. All other variants exhibited a reduction of DmgH activity without gaining metformin hydrolase activity. For variant Thr$_{289}$Ser we determined a lower $K_M$ of 28 ± 2 mM and a slightly higher $k_{cat}$ of 1 s$^{-1}$ compared to the wild-type DmgH (Supplementary Fig. 15), reflecting the improved catalytic performance. However, no metformin hydrolase activity was observed for these variants. Thus, mutation Ser$_{184}$Cys that is close to the active site was introduced in the DmgHα variant with the whole stretch mutated (Fig. 3A and Supplementary Fig. 14). The dimethylguanidine hydrolase activity remained low, but comparable metformin hydrolase activity was observed (Fig. 3B). Thus, these four mutations could be the tipping point for the evolution of MefH from DmgH.

### Growth analysis of A. niigataensis DSM7050

In contrast to *A. niigataensis* MD1, *A. niigataensis* DSM7050 was not able to grow on metformin as the sole nitrogen, carbon, and energy source (Fig. 4)[18]. Our analyses of the recombinant proteins established that the tandem ureohydrolase genes of *A. niigataensis* DSM7050 encode DmgH and thus allow the hydrolysis of dimethylguanidine rather than metformin. To investigate if *A. niigataensis* DSM7050 could assimilate dimethylguanidine, the bacteria were incubated in minimal medium containing either 10 mM dimethylguanidine or 10 mM metformin as the only nitrogen, carbon and energy source. As control, the minimal medium was supplemented with 10 mM ammonium chloride and 0.5% w/v glycerol. Cultures were incubated at 30 °C and OD$_{600}$ was recorded to assess growth. *A. niigataensis* DSM7050 grew on dimethylguanidine with a doubling time (t$_D$) of 12.9 h, corresponding to approximately half the growth rate of the control culture with ammonium and glycerol (t$_D$ = 6.4 h).

### Discussion

Bacteria able to use metformin as sole nitrogen, carbon and energy source were found to harbor an operon containing a nickel chaperon system and two consecutive ureohydrolase family genes[18-20]. We showed that the two ureohydrolase family proteins formed an α$_2$β$_4$ heterohexamer, a finding unprecedented in the ureohydrolase family (Fig. 2). The activity of this metformin hydrolase is also unusual because it produced dimethylamine and guanylurea rather than releasing urea (Fig. 1). In contrast, we found that the homologous enzyme from *A. niigataensis* DSM7050, which was not able to assimilate metformin[18] (Fig. 4), catalyzed the hydrolysis of dimethylguanidine (Fig. 1). This reaction represents a more typical reaction for ureohydrolase family proteins, as it yields dimethylamine and urea. Together with urease, which is encoded downstream of the investigated gene cluster, and previously described genes for C1 metabolism[18,19], the expression of dimethylguanidine hydrolase enabled the bacteria to grow on dimethylguanidine as sole carbon, energy, and nitrogen source (Figs. 1 and 3). Similar to metformin for *A. niigataensis* MD1[18], dimethylguanidine was a poor substrate for *A. niigataensis* DSM7050 and the doubling time was substantially longer compared to growth on glycerol and ammonium (Fig. 4).

The crystal structure revealed that MefH is a dimer of trimers as expected for a member of the ureohydrolase family[25]. However, the trimer is formed by one MefHα subunit that is framed by two MefHβ subunits. The crystal structures of Tassoulas et al. and Zhou et al. also contain six subunits with the same composition and orientation in the asymmetric unit[22,23]. Zhou et al. support the interpretation as a hexamer with state-of-the-art SEC-MALS data of the protein complex in solution[23]. The MefHβ and DmgHβ subunits are degenerated ureohydrolase family proteins, because four of the six residues that usually

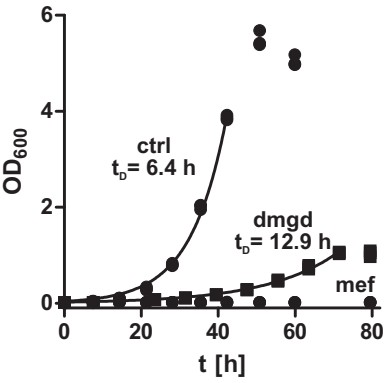

**Fig. 4 | Growth of *A. niigataensis* on different carbon and nitrogen sources.**
*A. niigataensis* DSM7050 was grown at 30 °C in a minimal medium containing 10 mM ammonium and 0.5% (w/v) glycerol (ctrl.), 10 mM dimethylguanidine (dmgd) or 10 mM metformin (mef) as carbon, nitrogen, and energy source. OD600 was measured to assess bacterial growth. Data points during exponential phase were fitted with an exponential equation to calculate the doubling time $t_D$. Data are from three independent cultures. Source data are provided as a Source Data file.

coordinate the two metal ions in the active center are mutated and the crystal structure showed that the β subunits do not contain metal ions (Fig. 2D and Supplementary Fig. 4). In contrast, each MefHα subunit formed a catalytic center with two metal ions typical for the ureohydrolase family. Subunit swapping confirmed that MefHα and DmgHα are mediating the substrate specificity of the active complexes (Fig. 3B). However, MefHα and DmgHα were not active when expressed alone, suggesting that the second subunits are interchangeable but essential for the formation of the active complexes. As previously described for guanidine hydrolase GdmH[21], activity of MefH and DmgH was dependent on $Ni^{2+}$ instead of the more commonly observed $Mn^{2+}$ in the active sites. Activation of recombinant MefH and DmgH depended on the co-expression of the $Ni^{2+}$ chaperones HypA and HypB, which are encoded directly upstream of the two ureohydrolase genes. The nickel content of the purified recombinant enzyme determined by ICP-OES corresponds to 74% occupancy of the di-metal centers in the MefHα subunits, indicating that the specific activity of a fully activated enzyme may be even higher than the reported values (Fig. 1). The nickel dependency is further supported by reconstitution experiments of apo-enzyme, in which only $Ni^{2+}$ could fully activate the enzyme[22]. The different metal dependency of ureohydrolase family members remains an open question and is likely dependent on the host and the substrate of the respective enzyme[21,26,27]. The rather high $K_M$ values of MefH and DgmH match the properties of many members of the ureohydrolase protein family[21,28]. The catalytic efficiency of MefH and DmgH is rather low, but is high enough to support growth on metformin or dimethylguanidine, respectively (Fig. 4). Among the genes found to be essential for or induced by growth on metformin, several encode transport proteins. While members of the small multidrug resistance family were shown to export guanylurea[29], a codB-like transporter is a strong candidate for the putative uptake and accumulation of metformin into the bacterial cells[18–20]. Furthermore, growth of *P. mendocina* MET and the *A. niigataensis* strains on the respective substrate as carbon, nitrogen, and energy source supports the biological relevance of the enzyme systems despite their high $K_M$.

Although enzymatically active DmgHα and MefHα share 93.6% sequence identity (Supplementary Fig. 4), DmgH and MefH exhibit almost orthogonal substrate specificities (Fig. 1C). Mutation of $Ser_{288}$-$Thr_{289}$-$Ser_{290}$ in DmgHα to $Asn_{288}$-$Ser_{289}$-$Ala_{290}$ together with the $Ser_{184}Cys$ substitution impaired the substrate specificity and could represent the key mutations that led to the evolution of an efficient

metformin hydrolase. The purified $T_{289}S$ variant exhibited a higher specific activity (Fig. 3) but the consequence of this mutation under in vivo conditions remains to be determined. The high degree of similarity between MefHα and DmgHα hints at the relatively recent adaptation to metformin as substrate. Improved catalysis of DmgH by a single amino acid exchange suggests that also mutations obtained by enzyme engineering of MefH could result in substantially increased activity towards metformin hydrolysis. The phylogenetic analysis of MefH and DmgH homologs indicated that the change in substrate specificity occurred in the genus *Aminobacter* (Supplementary Fig. 12). The type strain *Aminobacter niigataensis* DSM7050, encoding DmgH in its genomic DNA, was isolated before 1992[30]. In contrast, the isolation of metformin degrading bacteria was first reported in 2022[17–20]. Analysis of a homolog with 70% sequence identity from *Hyphomicrobium* confirmed that the distant relative also hydrolyzes dimethylguanidine and not metformin (Supplementary Fig. 13). Mobilization of MefH on plasmids and the probable horizontal transfer to *Pseudomonas* strains has been described before[20]. Currently, it is unclear why *Aminobacter* strains and possibly other bacterial taxa have developed a specific enzyme system for the degradation of dimethylguanidine, as this compound has not yet been identified as a guanidine compound present in biological systems. Nevertheless, the clustering of DmgH with urease genes supports the notion that dimethylguanidine is indeed the native substrate. The presence of nearly identical MefH genes and proteins in all metformin-degrading bacteria identified so far indicates a recent neo-functionalization, presumably of a DmgH precursor, as an adaptation to the massive synthetic production and medical application by humans.

## Methods
### Bacterial strains, cultivation and growth analysis
*E. coli* BL21(λDE3) gold (Invitrogen) was routinely grown in LB with 50 μg/ml kanamycin when transformed at 37 °C. *A. niigataensis* DSM7050 type strain was obtained from the German collection of microorganisms and cell cultures (DSMZ). It was routinely grown on nutrient agar at 30 °C. For growth assays, *A. niigataensis* DSM7050 was grown at 30 °C with 210 rpm in test tubes in 3 mL minimal medium (8.5 g/L $Na_2HPO_4$·$2H_2O$, 3 g/L $KH_2PO_4$, 0.5 g/L NaCl, 2 mM $MgCl_2$, 100 μM $CaCl_2$) that was supplemented with trace elements (0.1 mM EDTA, 0.03 mM $FeCl_3$, 6.2 μM $ZnCl_2$, 0.76 μM $CuCl_2$, 0.42 μM $CoCl_2$, 1.62 μM $H_3BO_3$; 0.08 μM $MnCl_2$) and vitamins (0.1 mg/L cyanocobalamin, 0.08 mg/L 4-aminobenzoic acid, 0.02 mg/L D-(+)-biotin, 0.2 mg/L niacin, 0.1 mg/L Ca-D-(+)-pantothenic acid, 0.3 mg/L pyridoxamine hydrochloride, 0.2 mg/L thiamine hydrochoride) supplemented with dimethylguanidine and metformin. All substrates were added to a final concentration of 10 mM. As positive control the minimal medium was supplemented with 0.5% (w/v) glycerol/10 mM $NH_4Cl$. $OD_{600}$ was recorded in cuvettes with 1 cm path length with a photometer (Eppendorf). Data points in exponential phase were fitted with an exponential equation in GraphPad Prism to establish the doubling time.

### Cloning, protein overexpression and purification
The *dmgH* operon was amplified from *A. niigataensis* DSM7050 genomic DNA. Operons from *A. niigataensis* MD1, *P. mendocina* MET and *Hyphomicrobium sp.* isolate SMAG_U6089 were purchased as DNA fragments from Twist Bioscience. Oligonucleotides used in this study can be found in Supplementary Data 1. The DNA fragments and PCR product were cloned into a pET24 vector derivative by Gibson assembly. A Strep-tag followed by a TEV cleavage site was introduced by whole plasmid PCR to obtain N-terminally tagged versions of the first of the ureohydrolase family proteins. For recombinant protein expression, *E. coli* BL21(λDE3) gold (Invitrogen) was transformed with the expression construct, grown at 37 °C to an $OD_{600}$ of ~1.1, transferred to 18 °C and induced over night with 0.5 mM IPTG in the

presence of 0.5 mM $Ni^{2+}$ if not stated otherwise. The cells were harvested by centrifugation, resuspended in enzyme buffer (50 mM Tris-HCl pH 8, 100 mM NaCl) supplemented with 1x EDTA-free cOmplete protease inhibitor (Roche) and lysed by ultrasonication (Branson). Tris, where mentioned, was substituted with the same concentration of HEPES pH 8. After centrifugation at $12,000 \times g$ for 15 min at 4 °C, the soluble protein fraction was loaded onto gravity-flow NiNTA columns or onto 1 mL Strep-Tactin columns (IBA Lifesciences) with an ÄKTA start system (Cytiva). The columns were washed with enzyme buffer and subsequently eluted with either 500 mM imidazole or 50 mM biotin in enzyme buffer. The purified enzymes were desalted into enzyme buffer by passage through PD10 columns (GE Lifesciences) to remove imidazole and biotin. The concentration of the purified fractions was determined with Bradford assays using BSA as standard.

### Enzyme assays

5–15 μg of purified proteins were incubated at 30 °C for 1 h in enzyme buffer (50 mM Tris-HCl pH 8, 100 mM NaCl) with the suitable substrate in a total volume of 100 μl. If not stated differently, 2 mM substrate were used in the reaction. Under these conditions, DmgH and MefH mediated a linear production of urea from dimethylguanidine and guanylurea from metformin for at least 1 h, respectively (Supplementary Fig. 16). Reactions were quenched with 87% acetic acid for the urea colorimetric assay or 100% MeOH for LC-MS analysis of guanylurea and dimethylamine. Urea released by DmgH activity was determined colorimetrically as described previously[24]. 50 μl of enzyme assay was mixed with 100 μl of color reagent (62 mM butanedionmonoxime, 3.6 mM thiosemicarbazide) and 150 μl acid reagent (120 μM $FeCl_3$ and 10 mM phosphoric acid in 20% (w/v) sulfuric acid) and incubated for 10 min at 96 °C. Samples with known concentrations of urea were treated in the same way and the OD at 520 nm was used to establish a standard curve. Guanylurea standards were measured in the range of 10–500 μM in enzyme buffer. Metformin was added to obtain a combined concentration of 2 mM metformin and guanylurea. Thus the matrix of the standards for LC-MS measurements was similar to the enzyme assay samples. If concentrations of metformin higher than 2 mM were used in the enzyme assays, the samples were diluted with 80% MeOH in reaction buffer to adjust the concentration to 2 mM metformin. If the metformin concentration of the reaction was lower than 2 mM, metformin was added after quenching to the reaction to obtain a concentration of 2 mM, so that a similar composition of samples and standards was maintained. For LC-MS measurements, we used a Nucleodur HILIC column (250 mm length ×2 mm i.d., 3 μm particle size, Macherey-Nagel), which was equilibrated with buffer B (90% acetonitrile, 0.2% formic acid, 10 mM ammonium formate) and eluted with a linear gradient to 45% buffer A (10 mM ammonium formate, pH 3.0) over a 9 min period followed by an isocratic step with 45% buffer B for 8 min. The column was operated at 20 °C with a flow rate of 0.15 ml/min. Injection volume was 2 microliters. Retention times were 8.3 and 8.5 min for metformin and guanylurea respectively. MS detection was performed using single ion monitoring in positive ionization mode. The peak area for GU (m/z = +103) was determined and the GU concentration was plotted against the peak area (Supplementary Fig. 3). The data obtained from the standards was fitted with a second order polynomic equation (GraphPad Prism v. 5.01) that was used to calculate concentrations of GU in the enzyme assay samples.

### Crystallographic structure elucidation

Recombinant 6× His-$An$MefH was purified by Ni-affinity chromatography followed by size exclusion chromatography in 100 mM NaCl, 50 mM Tris-HCl pH 8 and concentrated to 13.3 mg/ml. The protein was subjected to crystallization trials using the vapor diffusion method at 18 °C on 96-well Intelliplates (Art Robbins Instruments) using sitting drops (400 nl total drop volume) containing equal volumes of protein and reservoir solutions. Crystals grew from 20% [w/v] PEG 3350, 100 mM Bis Tris Propane-HCl pH 6.5, 200 mM NaF.

For X-ray data collection, crystals were cryoprotected and liganded by soaking for 30 min in mother liquor supplemented with 30% (v/v) ethylene glycol and urea (at saturation) prior to flash vitrification in liquid $N_2$. X-ray diffraction data were collected on beamline P13 of the Deutsches Elektronen-Synchrotron (EMBL-Hamburg) under cryo-conditions (100 K) at a wavelength of 0.97626 Å. Data processing used the XDS/XSCALE suite[31]. Phasing was by molecular replacement in PHASER[32] using AlphaFold predicted models of each subunit as independent search models. The models were generated with AlphaFold[33] on the publicly available ColabFold server[34] and trimmed to remove N-terminal loops and sequence segments predicted with low reliability (pLDDT >70). Manual building was performed in COOT[35] and model refinement was carried out in Phenix.refine[36] applying isotropic B-factors, NCS restraints and TLS parameters (one TLS group per chain). The resulting structures of the individual $αβ_2$ trimers were virtually identical (with $RMSD_{NCS}$ values of 0.176 Å and 0.341 Å for α- and β-subunits, respectively). The final model was assessed using Molprobity[37]. Molecular images were rendered using the UCSF ChimeraX graphics system[38]. All phi/psi angles were in the allowed regions of the Ramachandran plot with 96.86% in favored regions. Further X-ray data statistics and model parameters are given in Supplementary Table 3.

### Size exclusion chromatography and dynamic light scattering

Size exclusion chromatography of $An$MefH was performed with a Superdex 200 10/300 GL column (GE Healthcare) in 100 mM NaCl, 50 mM HEPES pH 8. The protein size was calculated using the formula $V = -0.135 \ln(M_W) + 0.9717$ with $R^2 = 0.9813$ derived from analyzing protein standards (Thyroglobulin 669 kDa, Ferritin 440 kDa, Aldolase 158 kDa, Conalbumin 75 kDa, Ovalbumin 45 kDa). Particle size and size distribution of the peak fraction from the size exclusion chromatography was determined by dynamic light scattering using a Zetasizer Nano ZSP (Malvern Panalytical) and the software provided by the supplier.

### Metal content determination

Strep-$An$MefH was purified, desalted into enzyme buffer and subjected to inductively coupled plasma optical emission spectroscopy on an Agilent 5800 system. Metal standards (Agilent solution, 50 ppm) were diluted in enzyme buffer to obtain the same matrix. Standards and samples were measured in triplicates in axial mode. Manganese and nickel contents were determined at suitable wavelengths (257.610 nm and 231.604 nm) and IntelliQuant® was used to detect potential other components. Linear regression of the standards was used to determine metal contents with the Agilent ICP-OES Expert software (Supplementary Fig. 7).

### Sequence analysis

Multiple sequence alignments were constructed with Jalview[39] using Clustal Omega[40]. For assignment of the metal binding sites, the sequence of $Synechocystis$ sp PCC 6803 GdmH was used. For phylogenetic analysis, representative full-length hits with >24% sequence identity were selected from individual BLAST searches with MefHα and MefHβ in the NCBI non-redundant sequence database. The resulting alignment was manually trimmed in BioEdit[41] to eliminate all gaps. A phylogenetic tree was inferred with the web-interface of IQ-Tree[42] using default parameters and the tree was visualized with TreeViewer[43].

### Reporting summary

Further information on research design is available in the Nature Portfolio Reporting Summary linked to this article.

## Data availability

All data generated in this study are provided in the Source Data file. Model coordinates and experimental diffraction data have been deposited with the Protein Data Bank under accession code 8RYI. X-ray diffraction images are deposited at https://doi.org/10.5281/zenodo.10638641. Source data are provided with this paper.

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

## Acknowledgements
We acknowledge the proteomics center of the University of Konstanz for their support. We thank A. Joachimi and D. Galetskiy for technical assistance and the LC-MS measurements. We thank M. Marsiske for the introduction to the ICP-OES, H. Hilbert for the introduction to the DLS and T. Dorendorf for the analytical SEC measurement.

## Author contributions
M.S. and J.S.H. conceived the project. L.R., M.S. and D.F. performed the experiments. J.R.F., H.L., A.B. and O.M. performed protein crystallization, structure determination, analysis and modeling. M.S., D.F. and J.S.H. wrote the manuscript and prepared figures with input from all authors. The manuscript was reviewed and approved by all authors.

## Funding

## Competing interests
The authors declare no competing interests.
