## [Peer Review File · Nature Communications]

Metformin hydrolase is a recently evolved nickel-dependent heteromeric ureohydrolaseREVIEWER COMMENTS

Reviewer #1 (Remarks to the Author):

The authors describe the characterization of a heteromeric metforminase that is active as a hexamer composed as a centrosymmetric dimer of trimers. Interestingly, the genes encoding each subunit must be coexpressed in *E. coli* to produce active enzyme (no activity if genes are expressed individually and mixed). The authors show that enzymes from *Pseudomonas* and *Aminobacter* are highly specific for metformin and are Ni dependent, but have some activity against dimethylguanidine. Another enzyme from a related *Aminobacter* species that can't grow on metformin only had activity against dimethylguanidine. The authors nicely show that the alpha subunits mainly determine substrate specificity. Interestingly, the authors were able to pinpoint 4 residues that mutated from the progenitor enzyme to enable metformin utilization. X-ray crystallography was used to determine the quaternary structure of the enzyme. One of the subunits crystalized with biuret (or possibly guanylurea) outside the predicted active site.

The manuscript is well-written, the figures are effective, and the experimental work seems to be very high quality. I have only a few minor suggestions:

1. Line 56: Hillmann et al published the first genomes from metformin-utilizing *Pseudomonads* and should be cited (DOI: 10.1128/mra.00639-22)
2. Lines 64-44: Please edit this sentence for clarity.
3. Line 281: A study was recently published on metformin transport that would seem to be more appropriate to cite here (doi: 10.1101/2023.08.10.552832)
4. The authors propose that metforminase recently evolved from a more widespread progenitor enzyme that acts on dimethylguanidine. I believe this is true, but could more evidence be provided to back this up? For example, can the authors determine which year metforminase and dimethylguanidine sequences started to appear in metasequence data? If metforminase sequences only appear in the past few years, while dimethylguanidine sequences go back much further, that would seem to provide evidence that these genes recently evolved.
5. The presence of biuret (suggesting guanylurea could also bind) binding to a site other than the active site invokes the possibility that guanylurea may be an enzyme effector. Could the authors test for metforminase activity with increasing amounts of guanylurea, to test for product inhibition?
6. In general, I think the authors could do a more thorough job citing the recent literature on metformin and ureohydrolase family proteins. In addition to the two mentioned above, here are a few related references:

D. Funck et al., Discovery of a Ni²⁺-dependent guanidine hydrolase in bacteria. *Nature* 603, 515–521 (2022).

J. L. Wilkinson et al., Pharmaceutical pollution of the world's rivers. *Proc. Natl. Acad. Sci. U.S.A.* 119, e2113947119 (2022).

Reviewer #2 (Remarks to the Author):

The article entitled “Metformin hydrolase is a recently evolved, nickel-dependent, heterometric ureohydrolase” by M. Sinn et al. reports the initial biochemical characterization and crystal structure of metformin hydrolase. Metformin is a broadly used drug to treat diabetes, and its introduction to the environment has led to the identification of genes involved in its bacterial degradation as a carbon and nitrogen source. The topic of study is of interest for the recent adaptation of an arginase-like enzyme in hydrolysis of an anthropogenic molecule, and for the apparent Ni²⁺-dependency. This review does have some concerns that may be appeased through additional descriptions and/or experiments.

The author's use of genomic context, relating the presence of nearby homologous nickel transporters to support the nickel-dependence of metformin hydrolase, echoes their previous work on guanidine hydrolase. The experiments performed test the metal loading from coexpression with HypA/B that is bound and maintained through the purification scheme. The authors report purified protein samples maintained 80% nickel and 20% manganese using their current methods of introducing nickel metal to the bacterial cell cultures, suggesting that the enzyme may have a preference for binding manganese ions. Also, the metal content for expressions in the presence of the metal ions is not reported so we cannot know what is the loading capacity of these samples, and do they all have 20% manganese present. If no metal is loaded in those samples, then additional *in vitro* supplementation would be necessary to report on activity dependence of the metals, versus metal loading from purification. Previous experiments with some parasitic arginases saw maximal activity *in vitro* with up to 1 mM MnCl₂ supplementation. A cleaner experiment that removes metals through chelation treatment and then tests delivery of different metals *in vitro* with HypA/B/GTP (similar to what was tested in their previous work for guanidine hydrolase) may aid in determining if this is a Nickel-only cluster, or some other species that involves manganese. As discussed in the 2022 *Nature* article on the homologous guanidine hydrolase, the active site shows a strong resemblance to arginase enzymes, which have a seemingly identical metal coordination environment and overall protein fold and mostly bind a binuclear manganese center to perform hydrolysis, although there are a few reports of other metal ions. The reported *K_m* and turnover values for metformin hydrolase stand out and make this reviewer wonder if the metal site is not reconstituted correctly, and that the nickel transporters if active may be necessary for another role in the pathway, such as activation of urease to deal with the produced urea from this pathway (as one possible option). In that the *K_M* should be in the range of the concentrations for the substrate in the cell, this *K_M* would suggest a strong active transport of metformin into the cell toner or above 50 mM. Comparison to the reference given for arginase has a *K_M* < 10 mM for L-arginine, which is

found at higher concentrations.

Regarding the structural characterization of the enzyme, additional solution biophysical evidence for a homogenous population of the observed quaternary state from the crystal would nicely support the observed structure. As a newly identified quaternary configuration of alpha and beta subunits, more solution support and analysis would be helpful. Why would the alpha-only species be inactive? Does it form a higher order oligomer or behave as a monomer in solution without beta? The crystal structure data is of good quality. No electron density is reported for the enzyme and the active site residues, metal center, and urea ligand, making it not possible to assess the active site. Additional anomalous data supporting and testing which metal(s) is(are) bound are needed to distinguish between Ni²⁺ vs Mn²⁺ in the active site. Therefore, the authors must be clear about the ambiguity of what is being modeled in the active site without additional support. Are there any differences in coordination environment, etc., that the authors note which would further distinguish nickel ions bound versus manganese ions bound? Is it possible there is a mixture of both bound in the structure? Without the anomalous experiments, it is ambiguous. Such data would be of use to complement further biochemical experiments.

Some general notes from the manuscript and issues/questions with citations are below:

- Is DmgH also in *P. mendocina*?

Regarding Metals: Report quantified metals of purified enzymes used in biochemical assays? Perhaps the mutants simply haven't retained the metals during purification.

- metal analysis, report in μM metal and μM enzyme to report the amount of nickel per protein molecules found.

- it is interesting that Manganese was detected in the nickel-only samples. Suggests some affinity for Manganese. Some members of this family use manganese.

- KM values are all >50 mM, these are weak binders. Have they optimized uptake of these molecules?

Oligomer state: solution evidence for hexamer?

- label what is meant as 'cleft' between trimers

Figures:

Fig. 1F and 1G: no error bars appear to be drawn

Fig. 1D: Plotting different reactions on the same graph is confusing at first glance. Although fine, this reviewer wanted to mention this in case the authors would find a better way to distinguish the reactions.

Fig. 3A, the wording of showing side chains of the metal binding residues of DmgH in grey when the crystal structure of AnMefH is in grey is confusing. Shouldn't the alpha fold model be a different color from the crystal structure?

Fig. 3B, specific activity with S.D. to the hundredths place should have activity reported to the hundredths place.

SI Figure 8&9: show density for urea and biuret and the metal sites

- as the first crystal structure report, SI figures of representative electron for parts of the molecule, and the active site with and without metals, would be beneficial. A PDB submission report was not included with the submission, so this reviewer cannot assess the active site.

- the coordination geometry is not drawn. Does urea displace the presumed catalytic water?

In a later supplemental figure, the authors label which genes in the gene cluster are alpha and beta.

Perhaps this could be added to a main figure in the manuscript.

Page 3-

- Citing one reference of reducing mortality in COVID-19 patients without taking into consideration that many studies show no difference is disingenuous (ref 9).

- Citation 7 for use in polycystic ovary syndrome is misleading, as the reference states that at present, it is not recommended for treatment.

- Citations 10-12; it is okay to mention that there have been implications, but it needs to be clear that the book isn't out on metformin having an active role in these processes. Some references suggest it is indirect through regulation of metabolism.

- Citation 13: appears to not be the direct reference for metabolism results.

Page 4- 'Key experiments were also performed with the 6x His-tagged versions and yielded very similar results.' is rather vague.

Page 5- statistics for precision of plotted data in figure 5F would be good to have for Vmax etc.

Page 6- 'we used AnMefH as it shares a higher sequence identity with DmgH than PmMefH'. The similarity is mentioned later, but it would be better to include here.

- "The nickel content was 12 μmol (mg protein)⁻¹ and additionally 2 μmol (mg protein)⁻¹ manganese were detected, but no other transition metals (Supplementary Fig. 7). " —> would be helpful to put this in terms of the total amount of protein in the results.

- "The enzyme with the highest activity was obtained when expression occurred in the presence of Ni²⁺ with only residual activity for the other metal ions." It would be helpful to cite figure 1D here.

- "DmgH exhibited a slightly higher apparent temperature optimum than AnMefH and PmMefH around 56°C compared to 50°C, respectively (Fig. 1F)", update to Fig. 1G.

- "Enzymes were expressed in the presence of Ni²⁺ and purified by Strep-Tactin affinity chromatography. Purified enzymes were incubated with 0.16-100 mM of their preferred substrate (Fig. 1E)." references the incorrect figure. Update to Fig. 1F

Page 8: Do the authors mean DmgH or GdmH for the ancestor?

- when mixing separately purified subunits, is this from separate expressions of each or is this coexpression? Please clarify.

Reviewer #3 (Remarks to the Author):

The manuscript authored by Hartig and colleagues focuses on the bacterial degradation of metformin, an anthropogenic biguanidine derivative widely used in the therapeutic treatment of type 2 diabetes and recognised as one of the most commonly released pharmaceutical compounds worldwide. Bacterial degradation of metformin by aquatic and soil bacteria plays an important role in its environmental elimination. In recent years, metformin-degrading bacteria have been isolated and gene clusters and enzymes involved in metformin degradation have been identified. However, the first enzyme in the degradation pathway, responsible for the cleavage of metformin into guanylurea and dimethylamine, remains poorly understood.

In this manuscript, metformin hydrolase from an *Aminobacter* sp. was heterologously produced and extensively characterised for the first time kinetically, biochemically, structurally by X-ray crystallography. Identified as a di-nickel centre-containing heterohexameric enzyme (a₂b₄) belonging to the arginase protein family, it was shown that only the a-subunit carries a catalytic centre, with the enzyme showing activity only in the presence of both subunits.

Overall, the manuscript represents a comprehensive and robust body of work providing definitive evidence for the primary enzyme involved in metformin degradation. Given the widespread use and distribution of metformin, the importance of this research in the field is clear. Although the subunit composition of the hydrolase is unusual, the catalysed reaction is consistent with the characteristics of the well-studied ureido hydrolase family. The manuscript is well written and scientifically rigorous, contributing significantly to the advancement of knowledge in the field.

Major points:

One of the most interesting results of the paper is that a heterotrimer is mandatory for the activity of metformin hydrolase, although the b-subunits do not contain an active site. I think the authors could analyse the structure in more detail to answer the question of why the b-subunit is required, e.g. to form contact sites, increase overall stability. I am curious what would be the result if the a-subunit is modelled with alpha-fold in the absence of the b-subunit?

In the discussion, the authors may comment on the poor catalytic efficiency of metformin hydrolase. Is it an imperfect enzyme because it has only recently evolved? Are the authors sure that dimethylguanidine does not exist in nature? So what could be the function of the natural precursor?

Minor points:

L.21 replace 'bacteria' by 'species'

L.54 As guanylurea is the product, it is pretty clear what the mechanism will be, a typical ureo hydrolase-like hydrolysis. So, 'mechanism' should be replaced by 'enzyme'

L.102 is a repetition of line 95

L-140 It would be more informative to give the metal content in mol per mol enzyme

L.159 ..ureohydrolase family... is this the same as the arginase family, or an overarching super family?

L.168: Where does the urea bound come from: hydrolysis product of dimethylguanidin, or was it added to the buffer?

L. 174 Is there a difference of biuret bound/non-bound structure? In other words: could it be bound to an allosteric site? Does biuret effect the activity (positively or negatively)?

L.178: can the cleft between b and b' be marked in Fig. 4b?

L.192: I cannot follow why DmgH is the likely evolutionary precursor. Based on what observation?
Please specify.

Response to reviewers comments

We thank all reviewers for their positive feedback, their careful reading of our manuscript and the constructive criticism that helped us to further improve the manuscript. During the course of revision one study about metformin hydrolase was published in PNAS by Tassoulas et al. (<https://doi.org/10.1073/pnas.2312652121>) and one other study that is currently still under revision by Zhou et al. (<https://doi.org/10.21203/rs.3.rs-3656883/v1>) was made available on a preprint server shortly before our initial submission. While they largely confirm the results of our study regarding the metformin hydrolase, with dimethylguanidine hydrolase DmgH we uniquely present the putative evolutionary precursor of MefH.

REVIEWER COMMENTS

Reviewer #1 (Remarks to the Author):

The authors describe the characterization of a heteromeric metforminase that is active as a hexamer composed as a centrosymmetric dimer of trimers. Interestingly, the genes encoding each subunit must be coexpressed in *E. coli* to produce active enzyme (no activity if genes are expressed individually and mixed). The authors show that enzymes from *Pseudomonas* and *Aminobacter* are highly specific for metformin and are Ni dependent, but have some activity against dimethylguanidine. Another enzyme from a related *Aminobacter* species that can't grow on metformin only had activity against dimethylguanidine. The authors nicely show that the alpha subunits mainly determine substrate specificity. Interestingly, the authors were able to pinpoint 4 residues that mutated from the progenitor enzyme to enable metformin utilization. X-ray crystallography was used to determine the quaternary structure of the enzyme. One of the subunits crystalized with biuret (or possibly guanylyurea) outside the predicted active site.

The manuscript is well-written, the figures are effective, and the experimental work seems to be very high quality. I have only a few minor suggestions:

1. Line 56: Hillmann et al published the first genomes from metformin-utilizing *Pseudomonads* and should be cited (DOI: 10.1128/mra.00639-22)

Thank you for pointing out this study to us. We included it into our manuscript. (line 53)

2. Lines 64-44: Please edit this sentence for clarity.

We have split the sentence and reversed the order to increase clarity. (line 63-65)

3. Line 281: A study was recently published on metformin transport that would seem to be more appropriate to cite here (doi: 10.1101/2023.08.10.552832)

We included a sentence about guanylyurea exporters to shed light on this aspect of bacterial metformin metabolism. (line 309)

4. The authors propose that metforminase recently evolved from a more widespread progenitor enzyme that acts on dimethylguanidine. I believe this is true, but could more evidence be provided to back this up? For example, can the authors determine which year metforminase and dimethylguanidine sequences started to appear in metasequence data? If metforminase sequences only appear in the past few years, while dimethylguanidine sequences go back much further, that would seem to provide evidence that these genes recently evolved.

Thank you for raising these important and interesting questions. Bioinformatic analysis is hampered by the high degree of sequence identity between MefH and DmgH and the limited data about the determinants of substrate specificity: Although we have an initial hypothesis which residues are important for substrate specificity from the mutational analysis, the high degree of similarity between DmgH and MefH (>93% identity for the alpha-subunits) undermines the clear distinction of the two enzyme activities based on sequence comparison. Therefore, examining less closely related homologous should provide us with more insights into their ancestry. We expressed and purified a DmgH homolog from *Hyphomicrobium* (70 % seq. identity to DmgH α and 68% seq. identity to DmgH β). DmgH from *Hyphomicrobium* turned over dimethylguanidine but not metformin, supporting the hypothesis that DmgH is the evolutionary ancestor. We included these experiments as further evidence for the hypothesis that metformin degradation evolved recently from a dimethylguanidine-hydrolyzing ancestor.

5. The presence of biuret (suggesting guanylurea could also bind) binding to a site other than the active site invokes the possibility that guanylurea may be an enzyme effector. Could the authors test for metforminase activity with increasing amounts of guanylurea, to test for product inhibition?

We tested several methods to monitor the MefH reaction. Measuring the guanylurea produced in the reaction turned out to be most reliable. Therefore, assessing product inhibition with guanylurea would interfere with our quantification of enzyme activity. However, we tested the effect of biuret on enzyme activity, as also suggested by reviewer 3. The results are detailed in Response to reviewer 3 below. However, we did not observe significant allosteric regulation of enzyme activity by biuret.

6. In general, I think the authors could do a more thorough job citing the recent literature on metformin and ureohydrolase family proteins. In addition to the two mentioned above, here are a few related references:

D. Funck et al., Discovery of a Ni²⁺-dependent guanidine hydrolase in bacteria. *Nature* 603, 515–521 (2022).

J. L. Wilkinson et al., Pharmaceutical pollution of the world's rivers. *Proc. Natl. Acad. Sci. U.S.A.* 119, e2113947119 (2022).

Thank you for your suggestion. We now cite Wilkinson (2022) in the revised version of the manuscript (line 42). We included a statement about the metal dependency of ureohydrolase family proteins in the discussion, where we cite Funck (2022), Wang (2021) and Viator (2008) (line 304).

Reviewer #2 (Remarks to the Author):

The article entitled “Metformin hydrolase is a recently evolved, nickel-dependent, heterometric ureohydrolase” by M. Sinn et al. reports the initial biochemical characterization and crystal structure of metformin hydrolase. Metformin is a broadly used drug to treat diabetes, and its introduction to the environment has led to the identification of genes involved in its bacterial degradation as a carbon and nitrogen source. The topic of study is of interest for the recent adaptation of an arginase-like enzyme in hydrolysis of an anthropogenic molecule, and for the apparent Ni²⁺-dependency. This review does have some concerns that may be appeased through additional descriptions and/or experiments.

The author's use of genomic context, relating the presence of nearby homologous nickel transporters to support the nickel-dependent of metformin hydrolase, echoes their previous work on guanidine hydrolase. The experiments performed test the metal loading from coexpression with HypA/B that is bound and maintained through the purification scheme. The authors report purified proteins samples maintained 80% nickel and 20% manganese using their current methods of introducing nickel metal to

the bacterial cell cultures, suggesting that the enzyme may have a preference for binding manganese ions. Also, the metal content for expressions in the presence of the metal ions is not reported so we cannot know what is the loading capacity of these samples, and do they all have 20% manganese present.

Thank you for your careful reading. We included the Ni²⁺ concentrations of the medium in the caption of Fig.1 (line 122) and the methods section (line 361). We didn't determine the Mn²⁺ concentration in our medium, but it has been shown that *E. coli* expresses a proton-coupled high-capacity importer for Mn²⁺ (<https://onlinelibrary.wiley.com/doi/10.1046/j.1365-2958.2000.01774.x>).

If no metal is loaded in those samples, then additional in vitro supplementation would be necessary to report on activity dependence of the metals, versus metal loading from purification. Previous experiments with some parasitic arginases saw maximal activity in vitro with up to 1 mM MnCl₂ supplementation.

In our experiments, pre-incubation or supplementation of the assay with metal did not increase enzyme activity.

A cleaner experiment that removes metals through chelation treatment and then tests delivery of different metals in vitro with HypA/B/GTP (similar to what was tested in their previous work for guanidine hydrolase) may aid in determining if this is a Nickel-only cluster, or some other species that involves manganese. As discussed in the 2022 Nature article on the homologous guanidine hydrolase, the active site shows a strong resemblance to arginase enzymes, which have a seemingly identical metal coordination environment and overall protein fold and mostly bind a binuclear manganese center to perform hydrolysis, although there are a few reports of other metal ions.

The study of Tassoulas et al. (2024) that was published in PNAS after our submission describes experiments with apo enzymes obtained by stripping metals from the purified enzyme, supporting the nickel dependency of MefH. We included a reference to their results when discussing our metal dependency experiments. (line 302ff). See also the discussion about anomalous X-ray data below. In light of our own data, we included a statement that acknowledges the ambiguity of our own metal analysis (line 183).

The reported K_m and turnover values for metformin hydrolase stands out and makes this reviewer wonder if the metal site is not reconstituted correctly, and that the nickel transporters if active may be necessary for another role in the pathway, such as activation of urease to deal with the produced urea from this pathway (as one possible option). In that the K_M should be in the range of the concentrations for the substrate in the cell, this K_M would suggest a strong active transport of metformin into the cell toner or above 50 mM. Comparison to the reference given for arginase has a K_M < 10 mM for L-arginine, which is found at higher concentrations.

We agree that the K_M values reported by us in the initial version seem rather high. Tassoulas et al (2024) report that the metformin hydrolase was inhibited by Tris, which we used in all our experiments. We now repeated the kinetic characterization of AnMefH and AnDmgH and found that the K_M values are 20 to 30-fold lower in HEPES buffer, whereas k_{cat} was not affected by the buffer system. We included the data in the manuscript and also discuss the data presented in Tassoulas et al. (line 155-158, Supplementary Figure 8).

Regarding the structural characterization of the enzyme, additional solution biophysical evidence for a homogenous population of the observed quaternary state from the crystal would nicely support the observed structure. As a newly identified quaternary configuration of alpha and beta subunits, more solution support and analysis would be helpful.

In order to address the homogeneity of the protein complex, we now performed analytical size exclusion chromatography and subsequently subjected the enzyme preparation to dynamic light scattering (DLS). We found that the enzyme elutes at approximately 200 kDa (molecular weight of the heterohexamer calculated from the sequence is 236 kDa) and DLS shows a monodisperse peak at around 10 nm diameter that fits to the expected size of the heterohexamer as estimated from the crystal structure by measuring the molecular dimensions in ChimeraX. We have added these experiments to the manuscript (line 172ff, Supplementary Fig. 9). Zhou et al. (2024, preprint) determined the molecular weight of the active metformin hydrolase heterohexamer by size exclusion chromatography coupled with multi-angle light scattering to be 223.7 ± 0.6 kDa, confirming our conclusion.

Why would the alpha-only species be inactive? Does it form a higher order oligomer or behave as a monomer in solution without beta?

We can only speculate about these issues. We reason that the beta-subunit provides stabilizing effects in the complex. We expressed and tried to purify both subunits alone, but the alpha-subunit predominantly formed inclusion bodies during expression. We only achieved partial purification of the alpha-subunit and this preparation was catalytically inactive. In contrast, the beta-subunit could be purified in large quantities as a soluble protein. As expected, the purified beta subunit also had no catalytic activity. Tassoulas et al. (2024) were also not able to obtain soluble alpha-subunit alone, confirming our findings.

The crystal structure data is of good quality. No electron density is reported for the enzyme and the active site residues, metal center, and urea ligand, making it not possible to assess the active site.

The complete electron density map will be available together with the deduced structure in PDB. Figures showing the active site structure and the biuret ligand in overlay with the corresponding electron density maps are now provided as Supplemental Information that is referenced in the results section. (Supplementary Figure 10 and 11)

Additional anomalous data supporting and testing which metal(s) is(are) bound are needed to distinguish between Ni²⁺ vs Mn²⁺ in the active site. Therefore, the authors must be clear about the ambiguity of what is being modeled in the active site without additional support. Are there any differences in coordination environment, etc., that the authors note which would further distinguish nickel ions bound versus manganese ions bound? Is it possible there is a mixture of both bound in the structure? Without the anomalous experiments, it is ambiguous. Such data would be of use to complement further biochemical experiments.

The reviewer correctly states that the presence of manganese in the active site of the enzyme in the crystalline state cannot be excluded. Effectively, the analysis of the metal content in solution (reported in this manuscript) indicated that the metal center is largely occupied by nickel but that a small fraction of manganese ions is present. However, the quantitation of partial occupancies of metal ions in the crystal by means of X-ray anomalous data, as proposed, is complex in this case. This is because of the spectral characteristics of the anomalous signal of nickel and manganese, whose K-edges lie at 8.3328 keV (1.488 Å) and 6.539 keV (1.896 Å), respectively. Nickel still has anomalous diffraction below its K-edge, including the energy range at which the manganese K-edge occurs. Thus, the task at hand would require measuring the anomalous signal of manganese over the predictable background signal of the more abundant nickel. It is also the case that the crystals are sensitive to radiation damage and that radiation damage would predictably affect the strongly absorbing metal ions first and foremost, which would trouble the calculation of their occupancies. This could be more so in this case where the K-edge of manganese lies at a very low energy, with low X-ray energies reputedly causing more radiation damage per dose than higher energies. Overall, we agree that some information could be gained

through a carefully tailored multi-wavelength data collection. However, we estimate that in this specific case the effort to obtain reliable quantitative information is in no relation to the small amount of additional structure information regarding the metal occupancy. Accordingly, we have included a statement in the main text to credit the ambiguity of our modelling of the metal in the crystal structure (line 183).

Some general notes from the manuscript and issues/questions with citations are below:

- Is DmgH also in *P. mendocina*?

We did not find DmgH-like sequences in *P. mendocina* genomic DNA. As MefH is encoded on a plasmid in *P. mendocina* MET, we propose that it was acquired recently by horizontal gene transfer.

Regarding Metals: Report quantified metals of purified enzymes used in biochemical assays? Perhaps the mutants simply haven't retained the metals during purification.

The mutants can be expected to retain the metal as we did not interfere with the metal binding residues. The detection of catalytic activity in all mutants implies that metal binding took place, since the coordination of a hydroxide ion between two metal ions is the prerequisite for hydrolytic activity in all ureohydrolases characterized so far. Thus, it can be deduced that the metal binding site remained intact. The main purpose of the mutational study was the analysis of substrate specificity and, despite the sometimes low specific activities, it is clear that only the variant with four exchanged residues shows DmgH activity.

- metal analysis, report in μM metal and μM enzyme to report the amount of nickel per protein molecules found.

We changed the reported values according to your suggestion. (line 146)

- it is interesting that Manganese was detected in the nickel-only samples. Suggests some affinity for Manganese. Some members of this family use manganese.

We assume that because *E. coli* accumulates manganese (see responses to reviewer 1), it was found even in our enzymes expressed in the presence of nickel. We cannot rule out that manganese can be part of the active site. Therefore, we included a statement in the manuscript about the ambiguity. (line 183) However, we are convinced that the full activation of the enzyme is dependent on nickel, as shown by the expression of MefH in the presence of different metal ions. This notion is confirmed by reconstitution experiments of the metal-stripped enzyme reported by Tassoulas et al (2024).

- KM values are all >50 mM, these are weak binders. Have they optimized uptake of these molecules?

As stated in the responses to reviewer 1, this is due to sub-optimal assay conditions using Tris buffer. When we repeated the kinetic characterization replacing Tris with HEPES buffer, we established Km values roughly 20 to 30-fold lower. We assume that even under sub-saturating conditions in vivo, the reaction is fast enough to fuel bacterial growth, as observed for bacteria with metformin and dimethylguanidine as sole nitrogen source.

Oligomer state: solution evidence for hexamer?

As stated above, we performed size exclusion chromatography followed by DLS and have included these results in the manuscript (line 172ff). Furthermore, Zhou et al. (2024, preprint) determined the molecular weight of the active metformin hydrolase heterohexamer by size exclusion chromatography

coupled with multi-angle light scattering to be 223.7 ± 0.6 kDA, confirming our conclusion (Supplementary Fig. 9).

- label what is meant as 'cleft' between trimers

We added a label in Fig. 2B according to your suggestion.

Figures:

Fig. 1F and 1G: no error bars appear to be drawn

Thank you for your careful reading. We added the error bars. Please note that some error bars are so small that they are not visible. We added a statement to clarify the figure. (line 134)

Fig. 1D: Plotting different reactions on the same graph is confusing at first glance. Although fine, this reviewer wanted to mention this in case the authors would find a better way to distinguish the reactions.

We understand your point, and we tried to split the graph, which was not providing improved clarity in our opinion. Therefore, we decided to keep the graphs unchanged whereas we modified the Y-axis label to "GU or U" instead of "GU/U".

Fig. 3A, the wording of showing side chains of the metal binding residues of DmgH in grey when the crystal structure of AnMefH is in grey is confusing. Shouldn't the alpha fold model be a different color from the crystal structure?

Thanks for pointing out this mistake. The error was in the figure legend, which now correctly explains that the AnMefH structure (including the metal-coordinating side chains) is depicted in grey, whereas the AlphaFold model of DmgH is depicted in green. In the lower part of the picture, the two structures align so well that only one backbone is visible.

Fig. 3B, specific activity with S.D. to the hundredths place should have activity reported to the hundredths place.

We changed the table accordingly.

SI Figure 8&9: show density for urea and biuret and the metal sites - as the first crystal structure report, SI figures of representative electron for parts of the molecule, and the active site with and without metals, would be beneficial. A PDB submission report was not included with the submission, so this reviewer cannot assess the active site.

We apologize for the inconvenience. We now included the overlays of the structure with electron density maps and the final PDB validation report as supplementary files. Supplementary Figures 10 and 11.

- the coordination geometry is not drawn. Does urea displace the presumed catalytic water?

We included a detailed view of the active site with the metal coordination geometry as supplementary figure (Supplementary Fig. 10B). Additionally, we compared our crystal structure (subunit E) with that of Tassoulas et al. (PDB entry 8SP2, subunit J) which has a water molecule coordinated by the two nickel ions in one of the alpha subunits. In an overlay, the oxygen of the urea takes the position of the water oxygen (see Figure below). We did not include the comparison in the main manuscript as, despite being interesting, it does not contribute to the story line of our manuscript.

Figure 1 Comparison of the active site of the MefH crystal structure of Tassoulas et al. (2024) (PDB entry 8SP2, subunit J, cyan) with water bound (black sphere) and our structure (subunit E, red) with urea bound. Note that the two Ni²⁺ coordinate the urea oxygen and water, respectively. The oxygen atoms of urea and water are located at the same position in the active site, where water is expected to be bound for the nucleophilic attack on the substrate.

In a later supplemental figure, the authors label which genes in the gene cluster are alpha and beta. Perhaps this could be added to a main figure in the manuscript.

Thank you for your suggestion. We believe this makes the manuscript more comprehensible and have included the label in Figure 2A.

Page 3

- Citing one reference of reducing mortality in COVID-19 patients without taking into consideration that many studies show no difference is disingenuous (ref 9).

- Citation 7 for use in polycystic ovary syndrome is misleading, as the reference states that at present, it is not recommended for treatment.

- Citations 10-12; it is okay to mention that there have been implications, but it needs to be clear that the book isn't out on metformin having an active role in these processes. Some references suggest it is indirect through regulation of metabolism.

- Citation 13: appears to not be the direct reference for metabolism results.

Thank you for your reasonable criticism. As a critical review of metformin's potential as a "golden bullet" in various diseases is outside the scope of the manuscript, we have decided to remove the statements in question and replace them with a general statement that metformin is discussed to have beneficial effects in other diseases.

Regarding citation 13: We replaced citation 13 by the direct reference Graham (2011), now citation 7.

Page 4

- 'Key experiments were also performed with the 6x His-tagged versions and yielded very similar results.' is rather vague.

The statement is vague on purpose because the experiments with the His-tagged versions were not carried out in sufficient numbers of independent replicates to perform a statistically validated comparison. When it became clear that the enzyme is indeed Ni-dependent, it was obvious that purifying it via Ni-chelating NTA resins was not the best approach.

Page 5- statistics for precision of plotted data in figure 5F would be good to have for V_{max} etc.

We now report the R² values for the fits in the figure legend. (line 131)

Page 6- 'we used AnMefH as it shares a higher sequence identity with DmgH than PmMefH'. The similarity is mentioned later, but it would be better to include here.

The sequence identity is mentioned here now as well. (line 164)

- "The nickel content was 12 μmol (mg protein)⁻¹ and additionally 2 μmol (mg protein)⁻¹ manganese were detected, but no other transition metals (Supplementary Fig. 7). " → would be helpful to put this in terms of the total amount of protein in the results.

As also requested by reviewer 1, we now express the values as metal concentration per protein concentration. (line 146)

- "The enzyme with the highest activity was obtained when expression occurred in the presence of Ni²⁺ with only residual activity for the other metal ions." It would be helpful to cite figure 1D here.

Thank you for your careful reading, we cite figure 1D now.

- "DmgH exhibited a slightly higher apparent temperature optimum than AnMefH and PmMefH around 56°C compared to 50°C, respectively (Fig. 1F)", update to Fig. 1G.

Thank you for your careful reading. We updated the reference.

- "Enzymes were expressed in the presence of Ni²⁺ and purified by Strep-Tactin affinity chromatography. Purified enzymes were incubated with 0.16-100 mM of their preferred substrate (Fig. 1E)." references the incorrect figure. Update to Fig. 1F

Thank you for your careful reading. We updated the reference.

Page 8: Do the authors mean DmgH or GdmH for the ancestor?

We mean DmgH as the ancestor. GdmH (as the most similar ureohydrolase with experimentally determined function) was included in the phylogenetic tree to provide a root, we do not have reason to believe it is a direct evolutionary precursor of MefH or DmgH.

- when mixing separately purified subunits, is this from separate expressions of each or is this coexpression? Please clarify.

We rephrased the sentence that now states: "...we cloned constructs for the co-expression of MefHa with DmgHb and DmgHa with MefHb." (line 223)

Reviewer #3 (Remarks to the Author):

The manuscript authored by Hartig and colleagues focuses on the bacterial degradation of metformin, an anthropogenic biguanidine derivative widely used in the therapeutic treatment of type 2 diabetes and recognised as one of the most commonly released pharmaceutical compounds worldwide. Bacterial degradation of metformin by aquatic and soil bacteria plays an important role in its environmental elimination. In recent years, metformin-degrading bacteria have been isolated and gene clusters and enzymes involved in metformin degradation have been identified. However, the first enzyme in the degradation pathway, responsible for the cleavage of metformin into guanidylurea and dimethylamine, remains poorly understood.

In this manuscript, metformin hydrolase from an *Aminobacter* sp. was heterologously produced and extensively characterised for the first time kinetically, biochemically, structurally by X-ray crystallography. Identified as a di-nickel centre-containing heterohexameric enzyme (a2b4) belonging

to the arginase protein family, it was shown that only the α -subunit carries a catalytic centre, with the enzyme showing activity only in the presence of both subunits.

Overall, the manuscript represents a comprehensive and robust body of work providing definitive evidence for the primary enzyme involved in metformin degradation. Given the widespread use and distribution of metformin, the importance of this research in the field is clear. Although the subunit composition of the hydrolase is unusual, the catalysed reaction is consistent with the characteristics of the well-studied ureido hydrolase family. The manuscript is well written and scientifically rigorous, contributing significantly to the advancement of knowledge in the field.

Major points:

One of the most interesting results of the paper is that a heterotrimer is mandatory for the activity of metformin hydrolase, although the β -subunits do not contain an active site. I think the authors could analyse the structure in more detail to answer the question of why the β -subunit is required, e.g. to form contact sites, increase overall stability.

We included a SDS-PAA gel as stated above into the SI (Supplementary Fig. 5). The α -subunits tended to form inclusion bodies, as e.g. also illustrated by the SDS-PAA of the PmMefH metal dependency. Therefore, we reason that the β -subunit increases the solubility of the enzyme complex, but for now, we cannot give a final answer. Analysis of the interface between the subunits of the crystal structure did not provide additional insights.

I am curious what would be the result if the α -subunit is modelled with α -fold in the absence of the β -subunit?

[NOTE OM: I think what the reviewer is asking here is whether the α -subunit can fold stably as a stand-alone chain or whether its fold is dependent on the presence of the β -subunit, as it happens in an oligomer complex]

The α -subunit was modelled in AlphaFold as a stand-alone polypeptide chain and this yielded a model that approximated the final crystal structure for this subunit, hinting at the possibility that the α -subunit can fold correctly in the absence of the β -subunit, even when lacking stability. In fact, the AlphaFold model was used to phase the crystallographic data using molecular replacement. Subsequently, we modelled the α -subunits into the crystal structure of MefH to build a homotrimer of α -subunits. There were no obvious conflicts in this model that would explain the failure to obtain an α -only version of MefH in soluble form. As stated in the manuscript, the overall fold of the α and β subunits are very similar, except for the unstructured termini. The model did not reveal any meaningful differences. Therefore, it was not included in the manuscript.

In the discussion, the authors may comment on the poor catalytic efficiency of metformin hydrolase. Is it an imperfect enzyme because it has only recently evolved?

DmgH exhibited similar catalytic efficiencies as MefH, therefore the catalytic inefficiency seems to be a more intrinsic feature of the enzymes. We included a statement in the manuscript addressing the catalytic efficiency and putting it in context to the catalyzed reaction. (line 307) See also our responses above regarding the improved K_M values by using HEPES instead of Tris buffer.

Are the authors sure that dimethylguanidine does not exist in nature?

We are not sure at all; on the contrary, we believe that dimethylguanidine might well exist in nature. However, to our knowledge dimethylguanidine has never been reported to be produced or even exist in biological systems to date.

So what could be the function of the natural precursor?

We do not know yet, but being generally interested in the source and fate of guanidine and its derivatives in nature, this is definitely an interesting question that we hope to address in future endeavors.

Minor points:

L.21 replace 'bacteria' by 'species'

We exchanged bacteria to species.

L.54 As guanylylurea is the product, it is pretty clear what the mechanism will be, a typical ureohydrolase-like hydrolysis. So, 'mechanism' should be replaced by 'enzyme'

We rephrased the sentence accordingly.

L.102 is a repetition of line 95

We kept the sentences, as the latter specifies the reaction products and is needed as an argument for use of the urea colorimetric assay.

L-140 It would be more informative to give the metal content in mol per mol enzyme

We changed the text according to your suggestion. (line 146)

L.159 ..ureohydrolase family... is this the same as the arginase family, or an overarching super family?

Unfortunately, the two terms are used interchangeable. In our opinion, ureohydrolase family is more appropriate taking into account the diverse substrates of the enzyme family. "Arginase family" should only be used for the sub-family with arginine as preferred substrate.

L.168: Where does the urea bound come from: hydrolysis product of dimethylguanidin, or was it added to the buffer?

As stated in the manuscript the crystals were soaked with urea. (line 409) This treatment happened to yield the best diffraction data.

L. 174 Is there a difference of biuret bound/non-bound structure? In other words: could it be bound to an allosteric site? Does biuret effect the activity (positively or negatively)?

We investigated this possibility according to your suggestion. The specific activity exhibited a slight increase of 24% and 35% when 5 or 50 mM biuret was added to a reaction containing 5 mM metformin, respectively. However, due to the similar properties of biuret and guanylylurea, we cannot exclude the possibility that biuret influenced our established LC-MS analysis of the reaction, resulting in an overestimation of guanylylurea. Furthermore, as shown in the new supplementary figure 11A, the biuret binding site is far away from the active site cavity. Additionally, an overlay of the biuret binding site of our crystal structure and that of Tassoulas et al. (2024) (PDB entry 8SNF) revealed no structural differences extending beyond the immediate vicinity of the ligand (see below). In summary of these consideration, we presently assume that biuret (or guanylylurea) has no pronounced allosteric effect on the activity of MefH.

L.178: can the cleft between b and b' be marked in Fig. 4b?

As also requested by reviewer 2, we added a label in Fig. 2B according to your suggestion.

L.192: I cannot follow why DmgH is the likely evolutionary precursor. Based on what observation? Please specify.

We have extended the phylogenetic tree in Figure S12 to include the sequences of MefH from all six bacterial isolates that were reported to degrade metformin (with >97% sequence identity among each other). The MefH α sequences form a single branch with *Aminobacter* DmgH as closest relative (93-94% sequence identity). Note that the MefH β sequences are identical in all six MefH enzymes. Additionally, we included the analysis of a further homologue of DmgH (70% sequence identity) (see also comment to reviewer 1). We show that it also exclusively hydrolyzed dimethylguanidine but not metformin (line 215ff). Given the high sequence similarity of *An*DmgH and MefH, we propose that metformin hydrolysis is a recent evolution of the dimethylguanidine hydrolase. The type strain *Aminobacter niigataensis* DSM7050 (containing DmgH in its genome) has been sampled before 1992, whereas the first reports on metformin-degrading isolates appeared in 2022. These either carry MefH genes in the genomic locus corresponding to DmgH in DSM7050 or on plasmids. In combination, we interpret these data as strong evidence for the evolution of MefH from DmgH in response to the anthropogenic release of metformin into the environment during the last 30 years.

REVIEWERS' COMMENTS

Reviewer #1 (Remarks to the Author):

The revised manuscript by the Hartig group addresses all of the concerns I brought up in my review. Guanlyurea wasn't tested for potential allosteric regulation, but there doesn't seem to be signs of product inhibition from the results in Fig 1F. It also seems unlikely that guanlyurea is an allosteric inhibitor since the substrate analog biuret was shown not have allosteric effects.

In general, I thought the authors did a good job of addressing other reviewers concerns. (e.g. by repeating their kinetic analysis with HEPES buffer instead of Tris). The issue of Ni vs Mn dependence may not be fully resolved, but the recent publications by Tassoulas et al. and Zhou et al. (preprint) also indicate Ni dependence, so it seems to be safe to assume Ni dependence.

The recent publications by Tassoulas et al. and Zhou et al. also address concerns about structure (and the necessity of the B subunit), as all three groups found that expressing both subunits simultaneously was necessary for activity, and all three groups determined similar structures (except that Tassoulas et al. determined a trimer and not a hexamer).

I have two minor points concerning the revision that I would like to see addressed:

1. The Hartig group determined the quaternary structure as a dimer of two heterotrimers (2 alpha and 4 beta subunits), and this is very similar to the structure determined by Zhou et al. However, Tassoulas et al. determined that the structure is a heterotrimer (1 alpha and 2 beta subunits). The structure of the heterotrimer is similar for all three groups, but there does seem to be evidence that multiple quaternary structures may exist. This should at least be brought up in the discussion.
2. I recommend revising the paragraph from lines 148-160. It seems illogical to report enzyme velocities determined with Tris buffer, then saying Tris buffer is an inhibitor, and use HEPES buffer to repeat the analysis for determining KM. I recommend reporting all values as determined with HEPES buffer. Then stating that Tris was also used but this impacted the KM, which is supported by Tassoulas et al. also showing Tris is an inhibitor.

Reviewer #2 (Remarks to the Author):

The authors of "Metformin hydrolase is a recently evolved, nickel-dependent, heterometric ureohydrolase" have submitted nicely performed additional experiments and manuscript updates based on reviewer comments. They have addressed the mentioned concerns of this reviewer. The authors additionally have noted that a PNAS article has since come out reporting the structure of

MfmAB from *P. mendocina* sp, which is being used to support the Ni-dependence experiments that were mentioned, and a preprint reporting the structure of MetCaCb. Interestingly, the MfmAB reports crystallization and solution data for a heterotrimer in comparison to the work presented here, and the MetCaCb structure reports a two alpha/four beta subunit complex. The reports of these three group's works should provide interesting comparisons to learn more about this relatively recently evolved system for metformin breakdown.

We would like to thank all reviewers for their constructive criticism of our study and manuscript. Please find our response to your final comments below.

Best regards

Jörg Hartig and Malte Sinn

Reviewer #1 (Remarks to the Author):

The revised manuscript by the Hartig group addresses all of the concerns I brought up in my review. Guanlyurea wasn't tested for potential allosteric regulation, but there doesn't seem to be signs of product inhibition from the results in Fig 1F. It also seems unlikely that guanlyurea is an allosteric inhibitor since the substrate analog biuret was shown not have allosteric effects.

In general, I thought the authors did a good job of addressing other reviewers concerns. (e.g. by repeating their kinetic analysis with HEPES buffer instead of Tris). The issue of Ni vs Mn dependence may not be fully resolved, but the recent publications by Tassoulas et al. and Zhou et al. (preprint) also indicate Ni dependence, so it seems to be safe to assume Ni dependence.

The recent publications by Tassoulas et al. and Zhou et al. also address concerns about structure (and the necessity of the B subunit), as all three groups found that expressing both subunits simultaneously was necessary for activity, and all three groups determined similar structures (except that Tassoulas et al. determined a trimer and not a hexamer).

I have two minor points concerning the revision that I would like to see addressed:

1. The Hartig group determined the quaternary structure as a dimer of two heterotrimers (2 alpha and 4 beta subunits), and this is very similar to the structure determined by Zhou et al. However, Tassoulas et al. determined that the structure is a heterotrimer (1 alpha and 2 beta subunits). The structure of the heterotrimer is similar for all three groups, but there does seem to be evidence that multiple quaternary structures may exist. This should at least be brought up in the discussion.

We have included a short paragraph in the Discussion section on the quaternary structures of the three studies. We are convinced that the biologically relevant enzyme form in solution is the heterohexamer as proposed by us and Zhou et al. This interpretation is supported by our own SEC and SLS data and is further supported by the state-of-the-art SEC-MALS data of Zhou et al. All three studies observed only a single peak in the SEC. The asymmetric unit of the crystal structure of Tassoulas et al. also contains six subunits with the same composition as in our and Zhou et al.'s structure. Their interpretation of MefH as a trimer is primarily based on their own SEC data, from which they estimated the Mw to be 179 ± 3 kDa, which is almost exactly in the middle of the Mws expected for a trimer or a hexamer. In our opinion, these data do not support the conclusion that the biological unit is a trimer.

2. I recommend revising the paragraph from lines 148-160. It seems illogical to report enzyme velocities determined with Tris buffer, then saying Tris buffer is an inhibitor, and use HEPES buffer to repeat the analysis for determining KM. I recommend reporting all values as determined with HEPES buffer. Then stating that Tris was also used but this impacted the KM, which is supported by Tassoulas et al. also showing Tris is an inhibitor.

We see the reviewers point and agree that it would read more logical. However, since other experiments like metal dependency and temperature dependency were conducted with Tris buffer, we feel that either readers could get the wrong impression that those experiments were conducted

also with HEPES or the text would become even more illogical and confusing. Therefore, we would like to refrain from the suggested revisions.

Reviewer #2 (Remarks to the Author):

The authors of “Metformin hydrolase is a recently evolved, nickel-dependent, heterometric ureohydrolase” have submitted nicely performed additional experiments and manuscript updates based on reviewer comments. They have addressed the mentioned concerns of this reviewer. The authors additionally have noted that a PNAS article has since come out reporting the structure of MfmAB from *P. mendocina* sp, which is being used to support the Ni-dependence experiments that were mentioned, and a preprint reporting the structure of MetCaCb. Interestingly, the MfmAB reports crystallization and solution data for a heterotrimer in comparison to the work presented here, and the MetCaCb structure reports a two alpha/four beta subunit complex. The reports of these three group’s works should provide interesting comparisons to learn more about this relatively recently evolved system for metformin breakdown.

Thank you for your positive feedback.